

# Transient features in charge fractionalization, local equilibration and non-equilibrium bosonization

Alexander Schneider[1], Mirco Milletari[2*] and Bernd Rosenow[1]

**1** Institut für Theoretische Physik, Universität Leipzig, D-04103 Leipzig, Germany
**2** Centre for Advanced 2D Materials and Department of Physics,
National University of Singapore, Singapore, 117551

* milletari@gmail.com

## Abstract

In quantum Hall edge states and in other one-dimensional interacting systems, charge fractionalization can occur due to the fact that an injected charge pulse decomposes into eigenmodes propagating at different velocities. If the original charge pulse has some spatial width due to injection with a given source-drain voltage, a finite time is needed until the separation between the fractionalized pulses is larger than their width. In the formalism of non-equilibrium bosonization, the above physics is reflected in the separation of initially overlapping square pulses in the effective scattering phase. When expressing the single particle Green's function as a functional determinant of counting operators containing the scattering phase, the time evolution of charge fractionalization is mathematically described by functional determinants with overlapping pulses. We develop a framework for the evaluation of such determinants, describe the system's equilibration dynamics, and compare our theoretical results with recent experimental findings.



# 1 Introduction

Interactions play a major role in the physics of one-dimensional systems. Due to the reduced dimensionality, excitations in one dimension can only occur collectively, and the system is therefore strongly correlated. As a consequence, the quasiparticle concept of Fermi liquid theory does not apply and these systems are better understood in the framework of Luttinger liquid theory, where for instance the quantum critical behaviour of the system is captured by non-trivial power law exponents [1–3]. A convenient way to study Luttinger liquids is through the method of bosonization, where a system of interacting fermions is related to an equivalent system of non-interacting bosons. This remarkable identity allows one to evaluate fermionic correlation functions exactly for the important case of forward scattering interactions, where the peculiar phenomena of spin charge separation is observed. From a mathematical point of view, the exact solution of the model is due to its integrability, i.e. the existence of an infinite number of conserved quantities that in turn precludes the systems from global equilibration. This "equilibrium bosonization" framework has been very useful in the past years to study systems such as carbon nanotubes, polymers, quantum wires or quantum Hall edge states [4–8].

Recently, there has been much interest, both experimentally [9–11] and theoretically [12–30], in the study of one-dimensional (1d) electron systems out of equilibrium. One obvious reason concerns the above mentioned integrability of these systems and therefore the understanding of whether or not equilibration can ever be reached. In the past few years it has become clear that in integrable systems some type of relaxation occurs, even though not towards the Gibbs equilibrium ensemble [32]. Remarkably, the necessity of correctly taking into account some particular non-equilibrium configurations, also revealed the necessity of modifying the standard bosonization approach. Indeed, it was found [14] that the Dzyaloshinskii-Larkin theorem [2], that ensures the non-interacting nature of the associated bosonic theory, does not hold out of equilibrium. Nevertheless, even in this case, the recently developed non-equilibrium bosonization approach [14–19] provides a universal framework to study one-dimensional interacting systems by expressing fermionic Green's functions (GF) as functional determinants containing a characteristic scattering phase, similar to the problem of full counting statistics [33–35].

One particularly interesting aspect of non-equilibrium states in 1d electron systems is the occurrence of charge fractionalization [9,11,27,28,36–43]. In Refs. [27,28], non-equilibrium bosonization has been used to study shot noise in $\nu = 2$ quantum Hall edge states, which carry 1$d$ chiral fermions in two edge modes of different velocities [44,45]. Considering a non-equilibrium setup where edge mode 1 is biased but edge mode 2 is grounded, with short range interactions between the edge modes, it was shown that charge pulses injected into edge mode 1 decompose into eigenmodes of the composite edge [26–28,36]. As a consequence, oppositely and fractionally charged pulses travel with different velocities in edge mode 2 and a finite shot noise is measured in the neutral excitation channel [28]. In particular, in the long time limit (where pulses are well separated) the shot noise evaluated in Ref. [28] was found to be in good agreement with recent experimental findings [11]. An interesting question arising from Ref. [11] concerns the study of finite size effects or, in other words, the characterization of the full relaxation dynamics of the system and its consequences for the low frequency noise. In Ref. [29], finite size effects were studied for the interaction quench of a LL initially described by a double step distribution function, with emphasis on the decay of the quasi-particle weight and the power laws characterizing the asymptotic steady state.

In order to properly take into account finite size effects, in this work we provide a general framework to evaluate functional determinants of non-equilibrium bosonization numerically also in the short time limit, where the pulses overlap. In Sect. II, we first discuss the general framework for evaluating functional determinants for an arbitrary number of pulses and an

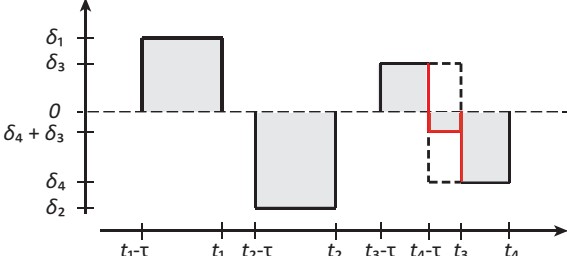

Figure 1: Schematic plot of a scattering phase $\delta_\tau(t)$ containing 4 window functions $w_\tau(t, t_j)$ of width $\tau$ and amplitudes $\delta_j$. The first two pulses on the left are fully separated in time, while the other two pulses on the right have a finite overlap (dashed lines). Through a rearrangement of the window functions $w_\tau(t, t_3)$ and $w_\tau(t, t_4)$ (red lines) describing the overlapping pulses, five non-overlapping time windows are obtained.

arbitrary separation between them. We show that in the overlapping regime a pure quantum treatment of the problem (as opposed to the semiclassical one of the non-overlapping regime) is essential in order to capture the relevant physical effects. In Sect. III, we apply our formalism to the problem of evaluating the equilibration process in a $\nu = 2$ quantum Hall system out of equilibrium during an interaction quench. The time dependence of the functional determinant is evaluated explicitly and put in correspondence with the relaxation dynamics of the system. In particluar, we clearly distinguish the regimes of quasi-particle creation and local equilibration, leading to a prethermalized, non-equilibrium steady state as described in Refs. [48, 49]. In Sect IV, consequences for the fractionalization noise are pointed out. In particular, we show how the relaxation of the initial non-equilibrium distribution function towards a steady state is related to the transition between a non-linear (in the external bias) and a linear shot noise characteristic. In Sect. V, we compare our results with the experimental findings of Ref. [11] and we provide a new method for understanding those findings within the charge fractionalization model. We extract the internal interaction parameters from a description of the transient shot noise regime at low bias and determine the initial Fermi velocities of the edge channels.

## 2   Functional determinants containing overlapping pulses

Interacting Luttinger liquids out of equilibrium can be described by applying the non-equilibrium bosonization approach [15]. In this formalism, it is convenient to express the lesser (greater) GF $G^{<(>)}$ [46] as the product of the (equal position) zero temperature equilibrium GF and a normalized determinant, e.g. $G^{<(>)}(\tau) = G_0^{<(>)}(\tau)\bar{\Delta}_\tau(\delta)$, with

$$\bar{\Delta}_\tau(\delta) = \frac{\det\left[1 + (e^{-i\delta_\tau(t)} - 1)f(\epsilon)\right]}{\det\left[1 + (e^{-i\delta_\tau(t)} - 1)\theta(-\epsilon)\right]} . \tag{1}$$

Here, the time dependent scattering phase $\delta_\tau(t)$ contains information about the system's dynamics, while the "statistical" information is contained in the Fermi distribution function $f(\epsilon)$. The phase $\delta_\tau(t) = \sum_j w_\tau(t, t_j)\delta_j$ consists of several window functions, representing charge pulses, $w_\tau(t, t_j) = \theta(t_j - t) - \theta(t_j - \tau - t)$. The amplitude of the pulses $\delta_j = 2\pi\lambda_j$ contains information about the interactions in the system, with $\lambda_j = 1$ for a non-interacting system. We note that in equilibrium, Eq.(1) reduces to the finite temperature part of the fermionic GF, or it is simply equal to one at zero temperature [15]. Generally, the window functions in Eq.(1) are either well separated or they overlap depending on their width $\tau$ and their separation in time.

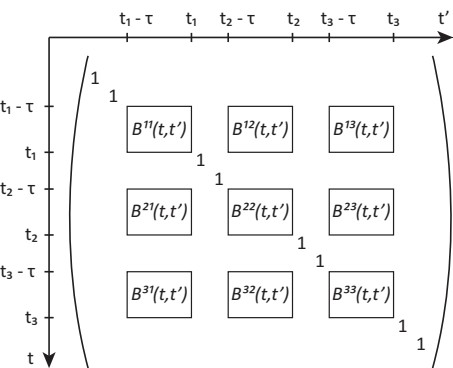

Figure 2: The matrix structure of $B(t, t')$ as defined in Eq. (7). Here, $B(t, t')$ is shown for a scattering phase containing 3 window functions.

However, it is always possible to combine and rearrange the overlapping window functions so that the phase $\delta_\tau(t)$ only consists of non-overlapping terms. Consider as an example the scattering phase for four pulses $\delta_\tau(t) = \sum_{j=1}^4 w_\tau(t, t_j)\delta_j$ with $\tau > |t_4 - t_3|$, see Fig. 1. The two overlapping window functions can be rewritten as

$$\delta_3 w_\tau(t, t_3) + \delta_4 w_\tau(t, t_4) = \delta_3 w(t, t_3 - \tau, t_4 - \tau) + \delta_4 w(t, t_3, t_4) + (\delta_3 + \delta_4)w(t, t_4 - \tau, t_3), \quad (2)$$

with $w(t, t_l, t_m) = \theta(t_m - t) - \theta(t_l - t)$. Using the projection property $w_\tau^2 = w_\tau$, it is possible to rewrite the term containing the non-overlapping window functions in Eq. (1) as [15]

$$(e^{-i\delta_\tau(t)} - 1) = \sum_j w_\tau(t, t_j)(e^{-i\delta_j} - 1) \equiv w_\tau^\delta(t, \{t_j\}). \quad (3)$$

In order to evaluate the energy and time dependent determinant defined in Eq. (1), $\hat{\epsilon}$ and $\hat{t}$ need to be treated as operators satisfying the commutation relation $[\hat{t}, \hat{\epsilon}] = i\hbar$. It is then convenient to define another projection operator $w_\tau^P(\hat{t}, \{t_j\}) = \sum_j w_\tau(\hat{t}, t_j)$ satisfying

$$\left[w_\tau^P(\hat{t}, \{t_j\})\right]^2 = w_\tau^P(\hat{t}, \{t_j\}), \quad (4)$$

$$w_\tau^P(\hat{t}, \{t_j\}) w_\tau^\delta(\hat{t}, \{t_j\}) = w_\tau^\delta(\hat{t}, \{t_j\}). \quad (5)$$

In this way, the numerator of Eq.(1) can be rewritten as

$$\det\left[1 + w_\tau^\delta(\hat{t}, \{t_j\})f(\hat{\epsilon})\right] = \det\left[1 + w_\tau^\delta(\hat{t}, \{t_j\})f(\hat{\epsilon})w_\tau^P(\hat{t}, \{t_j\})\right]. \quad (6)$$

In order to move the projector $w_\tau^P(\hat{t}, \{t_j\})$ to the right-hand side, we have rewritten the determinant as a trace over the logarithm, performed a series expansion of the logarithm and made use of the cyclic property of the trace. Using complete sets of eigenstates $\mathbb{1} = \int dt \, |t\rangle\langle t|$, $\mathbb{1} = \int d\epsilon \, |\epsilon\rangle\langle\epsilon|$ with the scalar product $\langle t|\epsilon\rangle = \frac{1}{\sqrt{2\pi}}e^{it\epsilon}$, the matrix elements in Eq.(6) can be reexpressed as

$$\langle t| \mathbb{1} + w_\tau^\delta(\hat{t}, \{t_j\})f(\hat{\epsilon})w_\tau^P(\hat{t}, \{t_j\})|t'\rangle = \delta(t - t') + w_\tau^\delta(t, \{t_j\}) \int d\epsilon \, \frac{e^{i\epsilon(t-t')}}{2\pi} f(\epsilon) \, w_\tau^P(t', \{t_j\})$$

$$= \delta(t - t')\left(1 - w_\tau^P(t, \{t_j\})w_\tau^P(t', \{t_j\})\right) + \sum_{l,m} B^{lm}(t, t')$$

$$\equiv B(t, t'). \quad (7)$$

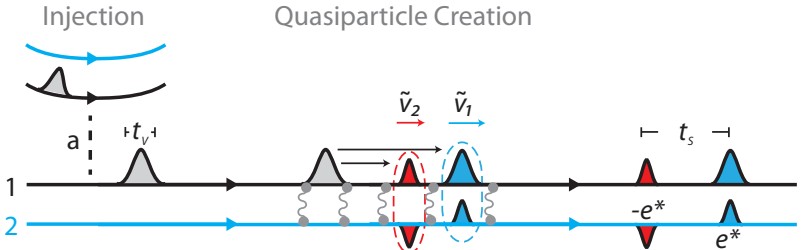

Figure 3: A charge pulse of time duration $t_v$ is injected into edge mode 1. Due to the interaction quench, the injected pulse decomposes into a charge and a neutral pulse, propagating with different velocities $\tilde{v}_i$. After the quench, two oppositely and fractionally charged pulses $e^*$ are separated in time by $t_s$ in edge mode 2.

Eq.(7) above is one of the main results of this work. The factors

$$B^{lm}(t, t') = w_\tau(t, t_l)\, b^l(t - t')\, w_\tau(t', t_m),$$
$$b^l(t - t') = \int d\epsilon\, \frac{e^{i\epsilon(t-t')}}{2\pi} \left(1 + (e^{-i\delta_l} - 1)f(\epsilon)\right) \tag{8}$$

are submatrices in the time domain, whose dimensions are determined by the widths of the respective time windows, see Fig. 2. In order to evaluate the above matrix elements, we discretize time and implement an energy cutoff procedure. We remark that due to the energy cutoff, a special regularization scheme is needed to perform the energy integration in the definition of the $b^l(t - t')$ and avoid unphysical Fermi edge singularities at the boundaries of the energy domain [20, 21]. We would like to emphasize that the determinant does not generally factorize into a product of diagonal blocks if not in the long time, semiclassical limit [16]. In the presence of overlapping pulses, the original full matrix structure needs to be considered in order to calculate the non-equilibrium GF over the full time-span $\tau$. Eqs. (7) and (8) are the general starting point of any particular numerical analysis.

# 3 Charge fractionalization and equilibration after interaction quench

In this section, we apply the general method described above to study the equilibration dynamics of two co-propagating integer quantum Hall edge modes, where one edge mode is brought out of equilibrium while the other one is kept unbiased. Therefore, fast charge pulses are injected into edge mode 1 with probability $a$, see Fig. 3. Non-equilibrium excitations are shared between the two edge modes due to interactions, which we assume to be turned on or off instantaneously by a quench. The two chiral modes are described by the Hamiltonian

$$\mathcal{H} = \hbar \int_x \left(v_1 \rho_1^2(x) + v_2 \rho_2^2(x) + v_{12}(t)\rho_1(x)\rho_2(x)\right), \tag{9}$$

where

$$v_{12}(t) = \begin{cases} v_{12} & \text{for } 0 < t \le t_Q \\ 0 & \text{else} \end{cases} \tag{10}$$

describes the particular interaction quench protocol in time. The quench in time can be realized by a spatial interaction region due to the system's chirality, see Sect. 4.

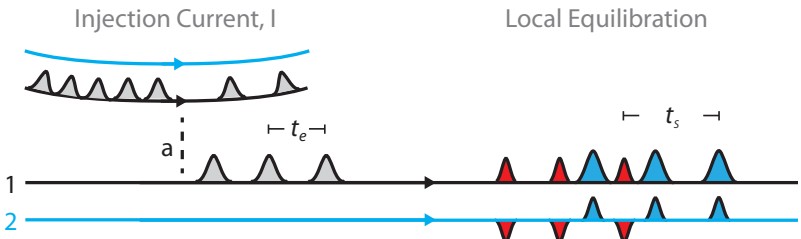

Figure 4: A random sequence of charge pulses is injected into edge mode 1, with an average time separation $t_e$ between them. After the interaction quench, the generated charge and neutral pulses mix among each other if $t_s > t_e$.

As a consequence of the interaction between upper and lower edge mode, a wave packet initially localized in edge mode 1 decomposes into a charge and a neutral pulse, which travel with different velocities $\tilde{v}_{1(2)} = v_{1(2)} \cos^2(\theta) + v_{2(1)} \sin^2(\theta) \pm \frac{1}{2} v_{12} \sin(2\theta)$ [28]. The relative interaction strength $\theta$ is parameterized by $\tan(2\theta) = v_{12}/(v_1 - v_2)$. The charge pulse consists of a charge $e^* = (e/2)\sin(2\theta)$ in edge mode 2 and a charge $e/2 + \sqrt{e^2/4 - (e^*)^2}$ in mode 1. The neutral pulse is composed of a charge $-e^*$ in edge mode 2 and a charge $e/2 - \sqrt{e^2/4 - (e^*)^2}$ in mode 1. Due to the different velocities, the centers of the charge and neutral pulse separate from each other until interactions are turned off and are finally separated in time by

$$t_s = \frac{v_{12}}{v_2 \sin 2\theta} t_Q , \tag{11}$$

see Fig. 3. Each pulse has a typical time width $t_v = \hbar/(eV)$ due to the injection with a given voltage $V$. Thus, depending on the relative magnitude of $t_s$ and $t_v$, the oppositely and fractionally charged pules in edge mode 2 are either well separated or do overlap. In addition, since a sequence of wave packets with an average time separation

$$t_e = t_v/a \tag{12}$$

is injected into edge mode 1, mixing of the charge and neutral pulses generated by different injected wave packets is possible, see Fig. 4. Hence, there are two processes taking place in edge mode 2: quasiparticle creation ($t_s > t_v$) and quasiparticle mixing ($t_s > t_e$).

In terms of the non-equilibrium bosonization approach, the above physics is captured by the effective scattering phase $\delta_\tau(t)$. The lesser GF $G_2^<(\tau) = i \langle \psi^\dagger(\tau)\psi(0)\rangle$ can be expressed as [15, 20, 21]

$$G_2^<(\tau) = G_0^<(\tau)\bar{\Delta}_\tau(\delta), \tag{13}$$

with[1]

$$G_0^<(\tau) = \frac{1}{2\pi} \frac{1}{(-i\tilde{v}_1 \tau + \alpha)^{\sin^2 \theta}} \frac{1}{(-i\tilde{v}_2 \tau + \alpha)^{\cos^2 \theta}}, \tag{14}$$

$$\bar{\Delta}_\tau(\delta) = \frac{\det\left[1 + (e^{-i\delta_\tau(t)} - 1)f_1(\epsilon)\right]}{\det\left[1 + (e^{-i\delta_\tau(t)} - 1)\theta(-\epsilon)\right]}, \tag{15}$$

where the scattering phase is found to be [28]

$$\delta_\tau(t) = \delta\{w_\tau(t, \tilde{t}_1) - w_\tau(t, \tilde{t}_2)\}. \tag{16}$$

---

[1]Actually, Eqs. (13) - (15) correspond to a snapshot at time $t_Q$ where interactions are still turned on (setup described in Sect. 4). Turning off interactions only effect the spectral properties, which are not discussed in Sect. 3.

Here, the amplitude $\delta = 2\pi(e^*/e)$ is related to the fractional charge, and $\tilde{t}_2 - \tilde{t}_1 = t_s$ denotes the time separation between two oppositely and fractionally charged pulses in edge mode 2. We describe the effect of preparing edge mode 1 with the injection of a random sequence of charge pulses in a non-interacting setting, and model the distribution function of edge mode 1 with a "double step" function

$$f_1(\epsilon) = a\theta(-\epsilon + \mu_1) + (1-a)\theta(-\epsilon + \mu_2) \,, \tag{17}$$

where $\mu_1 = (1-a)eV$ and $\mu_2 = -aeV$ are chosen symmetrically so that mode 1 carries a zero net current. In order to evaluate Eq.(15) over the full time-span $\tau$, the window functions need to be rearranged for $\tau > t_s$ as described in Sect. 2. Finally, the numerator of Eq. (15) can be expressed as the determinant of a $2 \times 2$ block matrix

$$\det\left[1 + (e^{-i\delta_\tau(t)} - 1)f(\epsilon)\right] = \det\begin{pmatrix} B^{11}(t,t') & B^{12}(t,t') \\ B^{21}(t,t') & B^{22}(t,t') \end{pmatrix}, \tag{18}$$

where the determinant is both over the block matrix structure and the discrete times. The block matrices $B^{lm}(t,t')$ can be expressed in terms of matrix elements $b^l(t-t')$, cf. Eq.(8), with

$$b^1(t-t') = \int_{-\Lambda}^{\Lambda} d\epsilon \, \frac{e^{i\epsilon(t-t')}}{2\pi} e^{-i\frac{\delta\epsilon}{2\Lambda}} \left(1 + (e^{-i\delta} - 1)f(\epsilon)\right),$$
$$b^2(t-t') = \bar{b}^1(t'-t), \tag{19}$$

where we introduced the additional phase factor $e^{-i\delta\epsilon/(2\Lambda)}$ to avoid jumps at the energy cutoff $\Lambda$ [15, 20, 21]. Due to the Toeplitz form of $B^{lm}(t,t')$ and due to the Hermitian relation between the matrix elements, the determinant is real. Inserting the "double step" function $f_1(\epsilon)$ and integrating over energy yields

$$b^1(t-t') \propto \frac{1}{\gamma_{t,t'}} (e^{-i\delta} - 1)\left(e^{i(1-a)eV\gamma_{t,t'}}a + e^{-iaeV\gamma_{t,t'}}(1-a)\right), \tag{20}$$

with $\gamma_{t,t'} = t - t' - \frac{\delta}{2\Lambda}$. At this point, we emphasize that all timescales $t_v$, $t_e$ and $t_s$ explicitly show up in Eq. (18): $1/t_v$ and $1/t_e$ set the frequency scale for oscillations of the matrix elements $b^1(t-t')$, and for times $t - t' > t_s$, the window functions part of $B^{lm}(t,t')$ are rearranged as described above.

In the limit of fully mixed quasiparticles ($t_s \gg t_e$), the off-diagonal submatricies are negligible and the determinant of Eq. (18) factorizes into the product of the two diagonal blocks [15, 28]. Then, Eq. (15) reads

$$\bar{\Delta}_\tau(\delta) \simeq \frac{\det[B^{11}(t,t')]\det[B^{22}(t,t')]}{\det[B_0^{11}(t,t')]\det[B_0^{22}(t,t')]} \equiv \bar{\Delta}_\tau^{\text{sep.}}(\delta) \,. \tag{21}$$

In this asymptotic (semiclassical) limit, the normalized determinant is found to decay exponentially towards zero [15]. However, this is not true in the opposite regime ($t_s \lesssim t_e$). Then, the full determinant $\bar{\Delta}_\tau(\delta)$ decays towards a non-zero asymptote. The asymptotic value is always determined by $\bar{\Delta}_{t_s}^{\text{sep}}(\delta)$ due to the rearrangement of the window functions at time $t_s$ and vanishing matrix entries $b^i(t-t')$ of the off-diagonal matrix blocks in the limit $\tau/t_s \to \infty$. This fact suggests that the normalized determinant including the off-diagonal matrix blocks can be approximated by a linear combination of the asymptote and an unknown $\tau$-dependent function

$$\bar{\Delta}_\tau(\delta) \approx \bar{\Delta}_{t_s}^{\text{sep}}(\delta) + (1 - \bar{\Delta}_{t_s}^{\text{sep}}(\delta))Y(\tau), \tag{22}$$

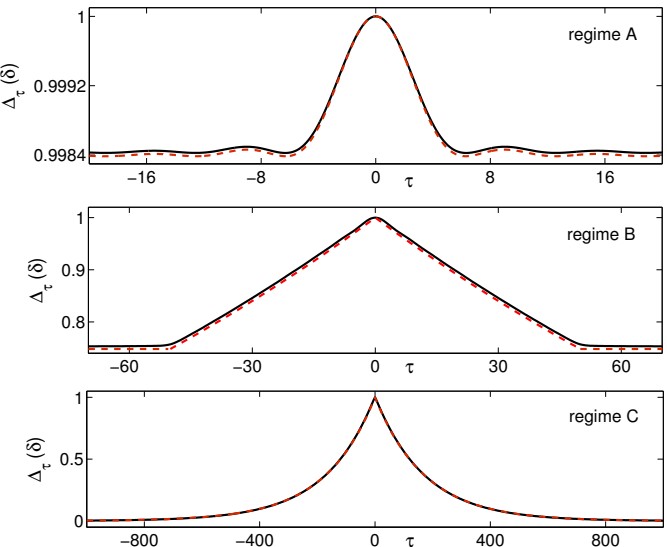

Figure 5: Normalized determinant $\bar{\Delta}_\tau(\delta)$ (black line) with $\theta = 0.47$ and $t_e/t_v = 100$ is plotted for $t_s/t_v = 1$ (regime A), 50 (regime B), 1000 (regime C), and compared to its analytic approximation (red dotted line). The asymptotic value is given by $\bar{\Delta}_{t_s}^{\text{sep.}}(\delta)$.

cf. Fig. 5. Furthermore, the asymptotic value $\bar{\Delta}_{t_s}^{\text{sep}}(\delta)$ defines the step height of the corresponding distribution function $f_2(\epsilon) \propto G^<(\epsilon)$. Keeping the normalization constraint $f_2(-\infty) = 1$ in mind, the distribution function is given by

$$f_2(\epsilon) = \int_\epsilon^\infty d\epsilon' \, \bar{\Delta}_{\epsilon'}(\delta) \,, \tag{23}$$

where $\bar{\Delta}_{\epsilon'}(\delta)$ is the Fourier transform of $\bar{\Delta}_\tau(\delta)$. In the following, we distinguish the following three regimes: regime A with $t_s \le t_v$, regime B with $t_v \ll t_s \ll t_e$, and regime C with $t_e \ll t_s$.

### 3.1 Regime A with $t_s \le t_v$

During the process of quasiparticle creation ($t_s \le t_v$), the asymptotic value is in good agreement with

$$\bar{\Delta}_{t_s}^{\text{sep}}(\delta) \approx 1 - \left(\frac{\delta}{2\pi}\right)^2 a(1-a)\left(\frac{t_s}{t_v}\right)^2 \,, \tag{24}$$

which can be obtained from a second order approximation of Eq. (20) together with a series expansion of Eq. (21). We find $Y(\tau)$ by its representation in Fourier space since $Y(\epsilon)$ assumes the form of a triangular function, which yields

$$Y(\tau) \approx \left(\frac{2}{eV\tau}\right)^2 \sin\left(\frac{eV\tau}{2}\right)^2 \tag{25}$$

after being transformed into time space again. The corresponding distribution function

$$f_{2u}(\epsilon) = \bar{\Delta}_{t_s}^{\text{sep}}(\delta)\,\theta(-\epsilon) +$$
$$\frac{1 - \bar{\Delta}_{t_s}^{\text{sep}}(\delta)}{2} \begin{cases} 2 - (\epsilon/eV + 1)^2 & -eV < \epsilon < 0 \\ (\epsilon/eV - 1)^2 & 0 \le \epsilon < eV \\ 2\,\theta(-\epsilon) & \text{else} \end{cases} \tag{26}$$

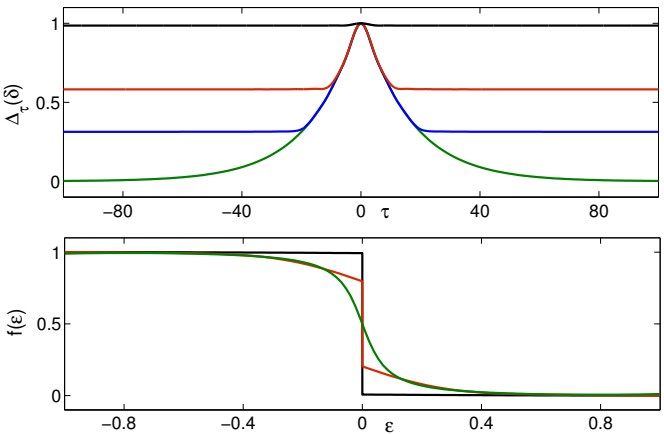

Figure 6: (top pannel) Normalized determinant $\bar{\Delta}_\tau(\delta)$ with $\theta = 0.47$ and $t_e/t_v = 10$ is plotted for $t_s/t_v = 1$ (black), 10 (red), 20 (blue) and 100 (green). (lower panel) The corresponding distribution function of mode 2 for $t_s/t_v = 1$ (black), 10 (red), 100 (green) evolves from an initial step function towards a smooth distribution function.

deviates quadratically from the initial step function. In the extreme case of $t_s \ll t_v$, almost no relaxation takes place $\bar{\Delta}_\tau(\delta) \approx 1$, and the GF of edge mode 2 does not deviate much from its equilibrium value, which is consistent with the picture of overlapping wave packets. If the time width of the wave packets is much larger than their separation in time ($t_v \gg t_s$), the oppositely and fractionally charged pulses of edge mode 2 "annihilate" each other after the interaction quench and the initially injected pulse is recaptured in edge mode 1. It thus looks as if the initial charge pulse freely propagated along edge mode 1 during the quench, leaving edge mode 2 in an unperturbed equilibrium state at zero temperature.

## 3.2 Regime B with $t_v \ll t_s \ll t_e$

In the regime of separated quasiparticles ($t_s > t_v$), the system locally equilibrates, see Fig. 5 (middle panel). We find analytically tractable results for $t_s \gg t_v$ and $a \ll 1$. In that case, the functional determinant can be derived from Eq. (21), which takes constant values for $\tau > t_s$ due to the rearrangement of the window functions. The determinant can be approximated by [15]

$$\bar{\Delta}_{t_s}^{\text{sep}}(\delta) \approx \exp\left(-\frac{t_s}{t_\phi}\right), \tag{27}$$

$$Y(\tau) \approx \frac{e^{(t_s-|\tau|)/t_\phi} - 1}{e^{t_s/t_\phi} - 1}\, \theta(t_s - |\tau|), \tag{28}$$

with $\pi\, t_\phi^{-1} = 2\, t_e^{-1} \sin^2(\delta/2)$. For $a \to 0$, the error at small time scales $|\tau| \lesssim t_v$ diminishes and the transition at $\tau = t_s$ gets sharp. The distribution function is found to be

$$
\begin{aligned}
f_2(\epsilon) = {}& e^{-\frac{t_s}{t_\phi}}\left(\theta(-\epsilon) + \frac{1}{\pi}\text{Si}(t_s\epsilon) - \frac{1}{2}\right) \\
& + \frac{1}{2} - \frac{1}{\pi}\text{Im}\big[\text{Ci}(t_s\epsilon + i t_s/t_\phi)\big] - \frac{1}{\pi}\text{Re}\big[\text{Si}(t_s\epsilon + i t_s/t_\phi)\big] \\
& + \frac{1}{2} - \frac{1}{\pi}\tan^{-1}(t_\phi\epsilon)
\end{aligned}
\tag{29}
$$

showing a non-trivial deviation from the initial step function.

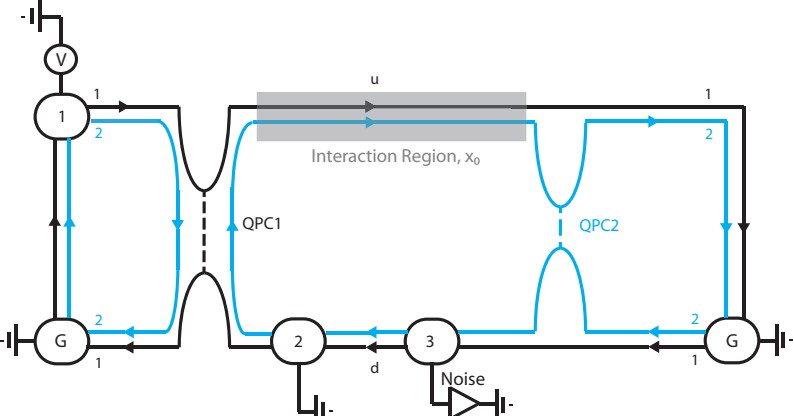

Figure 7: Sketch of a $\nu = 2$ Hall bar with a QPC1, where inner modes (2, light blue lines) are fully reflected, while partial transmission of outer modes (1, black lines) is possible. At a QPC2, the opposite situation is realized. The shaded area is the interaction region. The upper edge is biased with voltage V at contact 1; current noise is measured at contact 3.

### 3.3 Regime C with $t_s \gg t_e$

In the case of $t_s \gg t_e$, the functional determinant decays towards zero, see Fig. 5 (lower panel). For $a \ll 1$, the determinant can be approximated by its semiclassical representation

$$\bar{\Delta}_\tau(\delta) \approx \exp\left(-\frac{|\tau|}{t_\phi}\right), \tag{30}$$

with $\pi\, t_\phi^{-1} = 2\, t_e^{-1} \sin^2(\delta/2)$ neglecting errors at small time scales $|\tau| \lesssim t_s$. The corresponding distribution function is given by

$$f_2(\epsilon) = \frac{1}{2} - \frac{1}{\pi}\tan^{-1}(t_\phi \epsilon). \tag{31}$$

On a quantitative level, the system seems to reach a steady state for $t_s \gtrsim 10\, t_e$, which means that a few quasiparticle collisions cause a locally equilibrated steady state [49]. Thus, non-equilibrium excitations of edge mode 1 are fully shared with edge mode 2. As a consequence, the distribution function of edge mode 2 $f_2(\epsilon) \propto G_2^<(\epsilon)$ evolves from an initial step function ($t_s \ll t_v$) towards a smooth steady state ($t_s \gg t_e$), see Fig. 3.1.

## 4 Shot noise in the integer quantum Hall state

In order to measure the equilibration process discussed above in a $\nu = 2$ quantum Hall state with two integer edge modes co-propagating along the boundary of the system, the particular setup of Fig. 7 is suitable. The outer edge mode is labeled 1 and the inner one 2. The top and bottom edges originate at zero temperature from reservoirs at voltages $V_1 = V$ and $V_2 = 0$. There are two quantum point contacts (QPCs), which allow to partially backscatter edge currents. At QPC1, the outer modes are partially transmitted with probability $a$, while the inner ones are fully reflected; as a consequence, only the outer mode is noisy. Downstream of QPC1, the upper two edge modes interact over the finite distance $x_0$ (shaded area in Fig. 7) before reaching QPC2. Due to the chirality of quantum Hall edge modes, a finite interaction region in space is a realization of the interaction quench protocol described in the previous section.

At QPC2, the outer mode is fully transmitted while the inner one is partially reflected with probability $p$. The tunneling at QPC2 is described by

$$\mathscr{H}_{\text{QPC2}} = t_2 \psi_{2u}^\dagger(x)\psi_{2d}(x) + h.c.\,, \tag{32}$$

where the tunneling amplitude $t_2$ is related to the macroscopic tunneling probability $p$ via $p = |t_2|/(2\pi \tilde{v}_1^{2\sin^2\theta} \tilde{v}_2^{2\cos^2\theta})$ [28, 47]. Current noise in the partially reflected inner channel is then measured at contact 3. The low frequency shot noise can be expressed as

$$S_{\omega\to 0} = \frac{2e^2}{h} \frac{|t_2|^2}{2\pi} \int_\epsilon G_{2u}^<(\epsilon)G_{2d}^>(\epsilon) + G_{2d}^<(\epsilon)G_{2u}^>(\epsilon)\,, \tag{33}$$

where $\eta = u, d$ labels the upper and lower edge modes, and we have $G_{2\eta}^<(\epsilon) \propto f_{2\eta}(\epsilon)$ and $G_{2\eta}^>(\epsilon) \propto 1 - f_{2\eta}(\epsilon)$, which are proportional to the occupation of the electron states at QPC2. At zero temperature, the GFs of the lower edge are given by their equilibrium values e.g. $G_{2d}^<(\epsilon) = \theta(-\epsilon)/(\tilde{v}_1^{\sin^2\theta} \tilde{v}_2^{\cos^2\theta})$. The shot noise is therefore exclusively assigned to non-equilibrium effects in edge mode ($2u$) [11, 28], which are controlled by the three timescales $t_s$, $t_v$ and $t_e$. We note that in the quantum Hall setup

$$t_s = x_0(\tilde{v}_2^{-1} - \tilde{v}_1^{-1})\,. \tag{34}$$

In the following, we discuss how the different states of relaxation discussed in the previous sections are reflected in the dependence of shot noise on the bias voltage $V$.

*Regime A*: During the process of quasiparticle creation ($t_s \leq t_v$), the distribution function of edge mode ($2u$) deviates quadratically from the initial step function, and the shot noise is given by

$$S_{\omega\to 0} \propto a \sin(2\theta)^2 t_s{}^2 (eV)^3. \tag{35}$$

Its cubic dependence on $eV$ coincides with the results obtained in Refs. [27] and [26] in the Gaussian approximation.

*Regime B*: In the regime of separated quasiparticles ($t_s \gg t_v$), we use Eq. (27) and Eq. (28) to evaluate the resulting distribution function. We take the time dependence $\theta(t_s - |\tau|)$ in Eq. (28) into account by imposing a low energy cutoff at $1/t_s$ in the final energy integral. In addition, it is necessary to use a high energy cutoff since the approximation made in deriving Eq. (27) breaks down at small time scales. We choose $1/t_v$ as cutoff, which physically corresponds to the maximum energy range of the injected charge pulses. We find

$$S_{\omega\to 0} \propto aeV \log(t_s eV)\,. \tag{36}$$

*Regime C*: In the limit of fully mixed quasiparticles ($t_s \gg t_e$), the functional determinant decays towards zero and the low-energy cutoff $1/t_s$ gets irrelevant. One obtains for the shot noise generated at QPC2

$$S_{\omega\to 0} \propto a\, eV \log(1/a)\,. \tag{37}$$

Thus, the equlibration of edge mode ($2u$) is characterized by transitions between different power-laws in the shot noise signatures. Even if the transmission probability of QPC1 is $a \approx 1/2$, such that $a \ll 1$ is no longer satisfied, the shot noise obeys the cubic dependence $S_{\omega\to 0} \propto (eV)^3$ for $t_v \leq t_s$ and the linear dependence $S_{\omega\to 0} \propto eV$ for $t_s \gg t_e$, while the intermediary regime becomes blurred. However, the transient features of charge fractionalization are still apparent in the shot noise generated at QPC2, and can be calculated numerically exactly for all parameter values.

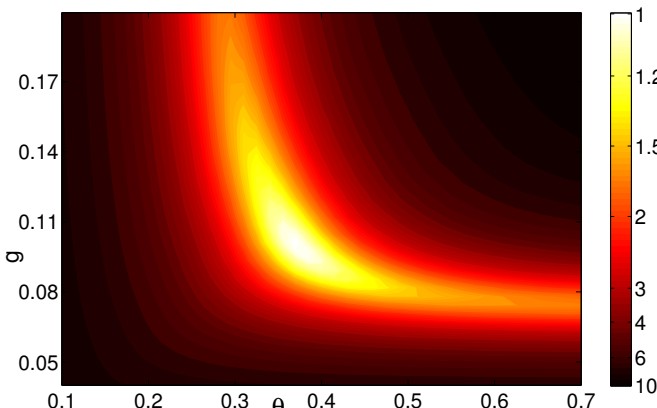

Figure 8: Relative fit error of the experimental data of Ref. [11] with regards to the theoretical description parameterized by the relative interaction strengths $\theta$ and $g$. The fit error is normalized with respect to the error found for the optimal fit parameters $\theta = 0.37 \pm 0.03$ and $g = 0.101 \pm 0.012$.

## 5 Comparison with experiments

We compute the shot noise as a function of the injected current for the setup described above, and compare it to the results of a recent experiment [11]. We take the finite length of the interaction region into account by including the off-diagonal matrix blocks of $\bar{\Delta}_\tau[\delta]$. Specifically, we consider the setup of Fig. 7 with an interaction distance $x_0 = 8\mu$m. The value $v_{12} = 4.6 \times 10^4$ m/s for the interaction strength is used as determined in the experiment Ref. [11]. There, shot noise measurements were performed as a function of the injected current $I = e^2/h$ for different values of $a$ and $p$. Here, we analyze the noise traces for four different tunneling probabilities $a = (0.51, 0.21, 0.86, 0.06)$ at QPC1. According to our model, there are two undetermined velocities, which we express via the independent parameters $\theta$ and $g = v_{12}^2/(4 v_1 v_2)$, with $0 < g < 1$. The latter boundary is motivated by the stability criterion of the Hamiltonian. We determine the two free parameters using a maximum likelihood plot based on the $\chi^2$ value for each parameter combination, which is a common procedure in numerical data analysis [50]. The relative fit error between the four numerical and experimental shot noise curves is shown in Fig. 8 for different pairs of $(\theta, g)$. Under the assumption of equally large error bars for different applied voltages, we find $\theta = 0.37 \pm 0.03$ and $g = 0.101 \pm 0.012$ to be the most likely values, and obtain from these the initial Fermi velocities $v_1 = (3.16 \pm 0.22) \times 10^5$ m/s and $v_2 = (2.65 \pm 0.16) \times 10^5$ m/s.

A slightly larger value of $\theta$ in the range $0.425 < \theta < 0.49$ was extracted in the experiment Ref. [11] by using the model of Ref. [28], which does not include the transient effects discussed in the present manuscript. Although the upper boundary $\theta \lesssim 0.4$ of the present analysis and the lower boundary $0.425 \lesssim \theta$ found in the analysis of Ref. [11] do not quite coincide, it is apparent from the likelihood plot Fig. 8 that for $g \approx 0.08$, there is a large region of $\theta$-values extending beyond $\theta \approx 0.45$, for which the relative fit error increases only very little, implying that the results of Ref. [11] are in very good agreement with the present analysis. It seems plausible that this difference can be accounted for by some residual dissipative mechanism at work in the experiment, beyond the coupling between the co-propagating edge states.

# 6 Conclusion

In this work we have developed a framework for numerically computing non-equilibrium functional determinants in a regime where individual pulses have overlap with each other, applicable for an arbitrary magnitude of the scattering phase. Our numerical approach is quite general and can be extended to many non-equilibrium problems, where the scattering phase consists of multiple window functions. We have illustrated our approach by applying it to the study of charge fractionalization in the $\nu = 2$ quantum Hall edge. We have characterized the transient regime of charge fractionalization according to the duration of the interaction quench with respect to time scales set by the injection energy of electrons and by the average time separation between injected electrons. In an experiment, the time over which the two edge modes interact is determined by the spatial distance between quantum point contacts. We have interpreted our results in terms of a semiclassical model of overlapping and mixing fractional charges, and have been able to extract microscopic model parameters by using the recent experimental results of Ref. [11].

# Acknowledgements

We acknowledge support from DFG Grant RO 2247/8-1. M. M. acknowledges support from the Singapore National Research Foundation under its fellowship program (NRF Award No. NRF-NRFF2012-01). We have benefited from helpful discussions with H. Inoue, M. Heiblum, I. Levkivskyi, and would like to thank H. Inoue, A. Grivnin, N. Ofek, I. Neder, M. Heiblum, V. Umansky, and D. Mahalu for providing the data used in Fig. 8.

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
