# Peer review of "Transient Features in Charge Fractionalization, Local Equilibration and Non-equilibrium Bosonization"

_SciPost Physics, doi:SciPost Phys. 2, 007 (2017)_

## Round 3 · Referee Report · Anonymous · 2017-1-14

Strengths

-

Weaknesses

-

Report

The analysis presented in this work is, to a large extent, of technical nature. I would like to commend the authors for being able to get their message(s) through, even for those readers who may not follow in detail each and every step of the derivation.

The authors’ two main thrusts are: (i) extend previous analyses of non-equilibrium bosonization. Specifically, the time evolution of injected charge is formally described by functional determinants with overlapping square pulses which describe the effective scattering phase. Unlike in the “semiclassical limit”, where those pulses are non-overlapping, here a full-fledged quantum mechanical treatment is required, and is developed by the authors. (ii) apply non-equilibrium bosonization (in this fully quantum limit) to the problem of electron fractionalization. Specifically, the authors address the transient behavior of fractionalization at the edge of \nu=2 integer QHE system, and compare their analysis to experimental data.

These are clearly significant contributions in this important field of non-equilibrium quantum electronic matter. As a further added value of this work, I would mention its clarity and the fact that the analysis is well embedded in a broad context of earlier works (on bosonization and on fractionaliation). I could easily see how the introduction of this paper is extended into a review of the field…

I therefore strongly recommend this work for publication.

Requested changes

Two items the authors ** may** want to consider (but should not constitute a condition for publication): First, in the introduction the authors refer to “recently developed non-equilibrium bosonization” and provide a list of references. In fac, non-equilibrium bosonization was first developed in Ref. 16 (and to some extent also in D.B. Gutman, et al, Phys. Rev.Lett. 2008) before the other references mentioned in the Introduction. Second, it would be intriguing to think how the results presented here should apply to more complex edges, e.g., the edge of a fractional bulk filling fraction (e.g., the $\nu=2/3$ FQHE), which has been recently studied experimentally.

  • validity: top
  • significance: high
  • originality: high
  • clarity: top
  • formatting: perfect
  • grammar: perfect

Author:  Mirco Milletari  on 2017-02-24  [id 104]

(in reply to Report 1 on 2017-01-14)
Category:
answer to question
suggestion for further work

We would like to thank the referee for his/her valuable comments and for recommending our paper for publication in SciPost. We are pleased to learn that the referee found our contribution “clearly significant”. Below we address the referee’s comments.

As for the First comment, we thank the referee for pointing to us the missing reference. We have modified the manuscript accordingly.

Concerning the Second comment, it would be indeed intriguing to think how our results generalise to more complex edges, e.g., the edge of a fractional bulk filling fraction (e.g., the ν=2/3 FQHE). Although the interferometer setup with edges at fractional filling fractions bear similarities with the problem considered in our paper, there are few important differences:

In our problem, we used the fact that the system before the first quantum point contact was non-interacting. Although this point seems crucial for the Quantum Quench model to work, it is possible to get around this restriction by using the original functional Keldysh approach developed in Ref. 16. We believe that this would allow to generalise our results to the case of Fractional filling fractions.
For the composite fractional edge, one would expect the existence of additional constraints on the dynamics of the system coming from its non-trivial topology. It would be certainly intriguing to understand the interplay of topology and strong non-equilibrium in this setup.

---

## Editorial Decision

published